# Amino Acid Composition in Various Types of Nucleic Acid-Binding Proteins

**DOI:** 10.3390/ijms22020922

**Published:** 2021-01-18

**Authors:** Martin Bartas, Jiří Červeň, Simona Guziurová, Kristyna Slychko, Petr Pečinka

**Affiliations:** Department of Biology and Ecology, Institute of Environmental Technologies, Faculty of Science, University of Ostrava, 710 00 Ostrava, Czech Republic; martin.bartas@osu.cz (M.B.); jiri.cerven@osu.cz (J.Č.); P19061@student.osu.cz (S.G.); p19071@student.osu.cz (K.S.)

**Keywords:** DNA, RNA, protein binding, G-quadruplex, triplex, i-motif, Z-DNA, Z-RNA, cruciform, amino acid composition

## Abstract

Nucleic acid-binding proteins are traditionally divided into two categories: With the ability to bind DNA or RNA. In the light of new knowledge, such categorizing should be overcome because a large proportion of proteins can bind both DNA and RNA. Another even more important features of nucleic acid-binding proteins are so-called sequence or structure specificities. Proteins able to bind nucleic acids in a sequence-specific manner usually contain one or more of the well-defined structural motifs (zinc-fingers, leucine zipper, helix-turn-helix, or helix-loop-helix). In contrast, many proteins do not recognize nucleic acid sequence but rather local DNA or RNA structures (G-quadruplexes, i-motifs, triplexes, cruciforms, left-handed DNA/RNA form, and others). Finally, there are also proteins recognizing both sequence and local structural properties of nucleic acids (e.g., famous tumor suppressor p53). In this mini-review, we aim to summarize current knowledge about the amino acid composition of various types of nucleic acid-binding proteins with a special focus on significant enrichment and/or depletion in each category.

## 1. Introduction

Interactions between proteins and nucleic acids (DNA and RNA) are central to all aspects of maintaining and accessing genetic information. Nucleic acid-binding proteins are mostly composed of at least one DNA or RNA-binding domain where the interfacing with amino acids takes place in a specific or nonspecific manner [1]. Identification of nucleic acid-binding proteins is one of the most important tasks in molecular biology. Currently, nucleic acid-binding proteins can be identified and further characterized by several experimental techniques, including pull-down assays [2,3], yeast one-hybrid system [4,5], electrophoretic mobility shift assays [6,7], chromatin immunoprecipitation [8,9], and by other specialized techniques [10,11]. However, it is time-consuming and expensive to identify nucleic acid-binding proteins by experimental approaches [12]. With the easy availability of a large amount of protein sequence data, there is a rapid development of computational approaches and prediction tools that can rapidly and reliably identify nucleic acid-binding proteins [13,14]. Several such tools model nucleic acid-binding abilities based on protein amino acid composition [15,16]. There is a growing interest in so-called noncanonical nucleic acid structures and proteins that preferentially bind them [17,18,19,20,21,22,23,24]. Noncanonical nucleic acid structures are DNA and RNA structures different from their basic form, i.e., double-stranded right-handed DNA or single-stranded RNA, and are often formed by simple nucleotide repeats [25,26,27,28]. Physiologically, they are represented mainly by G-quadruplexes [29], i-motifs [30], triplexes [31], R-loops [32], slipped hairpins [33], DNA cruciforms [34], RNA hairpins [35], and Z-DNA [36]. These DNA/RNA structures have important biological functions [37,38,39,40,41,42] and contribute to many human diseases [43,44,45,46]. It became more and more evident, that proteins preferentially interacting with these structures share distinct amino acid features/fingerprints [47,48]. This mini-review aims to focus on the amino acid composition of various types of DNA and RNA-binding proteins and to compare the amino acid composition of proteins that prefer binding to different noncanonical forms of nucleic acids.

## 2. Amino Acid Composition of Nucleic Acid-Binding Proteins

According to the Gene Ontology (GO) knowledgebase, there are 5037 nucleic acid-binding proteins (filtering GO:0003676 term by “protein”) with experimental evidence in *Homo sapiens* [49,50,51]. Of this number, 2572 are annotated as RNA-binding and 2439 as DNA-binding proteins (some proteins have both functions). 1768 human proteins are known to bind DNA in a sequence-specific manner. It would be interesting to quantify the overall amount of proteins binding nucleic acids in a structure-specific manner. Unfortunately, there is no such category yet. We strongly suggest revisions in this manner. Inspiration can be found in the following review papers/databases focused on specific properties of proteins binding to G-quadruplexes [19,52,53,54], cruciforms [55], and Z-DNA/Z-RNA [56].

### 2.1. History

Amino acid composition of some nucleic acid-binding proteins was intensively studied at the beginning of the 70s, when Koichi Iwai et al. determined that “calf-thymus histones comprise five main types which differ in amino acid composition and electrophoretic mobility: A glycine-rich, arginine-rich histone (also known as f2al or IV); a glutamic-acid-rich, arginine-rich histone (fe or III); a leucine-rich, intermediate type histone (f2a2 or IIb1); a serine-rich, slightly lysine-rich histone (f2b or IIb2); and an alanine-rich, very lysine-rich histone (f1 or I)” [57], by using specialized chromatographic technique followed by polyacrylamide gel electrophoresis. In 1975, from the comparison of 68 representative proteins and frequencies of 61 codons of the genetic code, it was found that the average amounts of lysine, aspartic acid, glutamic acid, and alanine are above the levels anticipated from the genetic code, and arginine, serine, leucine, cysteine, proline, and histidine are below such levels [58]. There are a couple of examples from the 90s and 2000s when amino acid substitution in nucleic acid-binding protein abolished its function, e.g., an arginine to lysine substitution in the bZIP (Basic Leucine Zipper) domain of an opaque-2 mutant in maize abolished specific DNA-binding [59], missense mutations (Met175Arg and Ser191Asn) abolishing DNA-binding of the osteoblast-specific transcription factor OSF2/CBFA1 in human patients with cleidocranial dysplasia [60], or impaired RNA-binding of fragile X mental retardation protein upon missense mutation IIe-304→Asn in one of its KH domain [61]. Recent advantages in sequencing and bioinformatic methods allow us to directly compare the amino acid composition of thousands of (not only) human nucleic acid-binding proteins [62,63]. One of the most popular programs for this purpose is, e.g., composition profiler [64], which is a web-based tool for semi-automatic discovery of enrichment or depletion of amino acids, either individually or grouped by their physicochemical or structural properties [64]. Scientists often find themselves in the situation when they only have a sequence of new “hypothetical” protein, derived mainly from transcriptome sequencing, and want to deduce its function [65]. In case that no meaningful alignment to protein with known function is available, there is still a way to get some useful information using only primary amino acid sequence and its composition. In 2003, Cai and Lin used a protein’s amino acid composition and support vector machine (SVM) prediction to decide if protein belongs to one of three classes—rRNA-, RNA-, or DNA-binding [66]. Currently, there are also user-friendly web-based prediction tools called DNAbinder and PseDNA-Pro, which can predict if the submitted protein sequence has DNA-binding ability [12,67].

### 2.2. Methods to Inspect the Amino Acid Composition of Proteins

Several approaches are used to inspect the amino acid composition of nucleic acid-binding proteins. Basically, we can divide the methods into in vitro and in silico. In vitro approaches are necessary to obtain a sequence of the protein of interest. Although the development of large-scale genomic sequencing has greatly simplified the procedure of determining the primary structures of proteins, the genomic sequences of many organisms are still unknown, and also modifications such as post-translational events (citrullination, deamidation, polyglutamylation,…) may prevent proper determination of the protein sequence [68]. Then, the complete characterization of the primary protein structure often requires a mass spectrometry method with minimal assistance from genomic data, i.e., de novo protein sequencing [68,69]. In silico approaches are based mostly on previous knowledge about primary protein sequence. There is currently a plentitude of bioinformatics tools designed for that purpose, see, e.g., [64,70,71,72,73].

### 2.3. Amino Acid Composition of Nucleic Acid-Binding Proteins

Nucleic acid-binding proteins are traditionally divided into two categories. The first category comprises proteins with the ability to bind DNA, and the second category comprises proteins that bind to RNA. This division is quite outdated, mainly because, from the historical perspective, proteins that bind RNA were typically considered as functionally distinct from proteins that bind DNA and studied independently. Interestingly, current gene ontology analyses reveal that DNA-binding is potentially a major function of the mRNA-binding proteins [74]. Nonetheless, several studies inspecting amino acid composition of DNA and/or RNA-binding proteins were published [75,76] and find that particular amino acid residues are generally enriched or depleted within these protein categories (see Table 1).

Another, even more important division of nucleic acid-binding proteins is based on a so-called sequence or structure-specific type of binding. Proteins able to bind nucleic acids in a sequence-specific manner usually contain one or more of the well-defined structural motifs. One of such motifs, zinc-finger, binds DNA (or RNA) through specific interaction with nucleotides and sugar-phosphate backbone. Tandem repeating of slightly different zinc-finger motifs in protein then allows to recognizing its consensus nucleic acid-binding sequence specifically. Cysteine and histidine amino acid residues are crucially important to coordinate Zn^2+^ binding in the largest and best-characterized subgroup of zinc-finger binding proteins named the Cys_2_His_2_ fold subgroup [77,78]. Other well-defined sequence-specific motifs—leucine zipper, helix-turn-helix, or helix-loop-helix—are listed in Table 1, together with their common signatures of amino acid residues.

In contrast, many proteins do not recognize nucleic acid sequence but rather local DNA or RNA structures (G-quadruplexes, i-motifs, triplexes, cruciforms, left-handed DNA/RNA form, and others) [19,30,41,55,90]. Finally, there are also proteins recognizing both sequence and local structural properties of nucleic acids (e.g., famous tumor suppressor p53 [91], Myc-associated zinc finger protein (MAZ) [92,93], and many RNA-binding proteins [94])—these proteins usually contain sequence-specific binding domain(s) together with domain(s)/region(s) with preference to noncanonical nucleic acid structures [95,96,97]. In 2016, Wang et al. analyzed the abundance of intrinsic disorder in the DNA- and RNA-binding proteins in over 1000 species from Eukaryota, Bacteria, and Archaea domains of life [98]. They have revealed a very interesting phenomenon that DNA-binding proteins had significantly increased disorder content and were significantly enriched in disordered domains in Eukaryotes but not in Archaea and Bacteria. The RNA-binding proteins were significantly enriched in the disordered domains in Bacteria, Archaea, and Eukaryota, while the overall abundance of disorder in these proteins was significantly increased in Bacteria, Archaea, animals, and fungi [98]. Disordered domains or regions are also extensively present in chromatin-binding proteins [99,100]. Interestingly, some disordered proteins or regions show very high structural specificity to the different types of noncanonical nucleic acids. For instance, human protein SRSF1 (Serine/arginine-rich splicing factor 1) contains several intrinsically disordered regions [101], which are compositionally enriched in glycine (14.11% of overall amino acid residues) and arginine (17.39% of overall amino acid residues) content. It was previously shown that SRSF1 has a high affinity to RNA G-quadruplex structure [102]. Subsequent analyses have shown that the dataset of 77 G-quadruplex binding proteins is significantly globally enriched in arginine, glycine, aspartic acid, asparagine, and valine, and depleted in cysteine and other amino acid residues [47] (Figure 1). Finally, the common amino acid motif in the form of RGRGRGRGGGSGGSGGRGRG was derived, and most of the currently known G-quadruplex binding proteins contain at least some modification of it [47]. Using this motif, a new dataset of G-quadruplex binding proteins was predicted from the set of all human DNA/RNA-binding proteins [47], and some of them were independently experimentally validated (e.g., CIRBP, which is a cold-inducible RNA-binding protein in the study by Huang and colleagues [103]). A similar study focused on an amino acid composition of cruciform binding proteins was also published, and the significant enrichment for lysine and serine amino acid residues has been revealed [48] (Figure 1). Unpublished results also indicate distinct amino acid profiles in Z-DNA/RNA and triplex binding proteins, both significantly enriched in aspartic acid and isoleucine and depleted in cysteine residues [89] (Figure 1). In future studies, it would be interesting to specifically analyze local amino acid composition (only in the nucleic acid interaction sites and their close neighborhood) in these proteins. Unfortunately for the vast majority of them, the knowledge about exact DNA/RNA binding site(s) is still missing.

As was shown above, proteins that preferentially recognize noncanonical nucleic acid structures often have a distinct amino acid composition with particular significant global enrichment and/or depletion of different amino acid residues. Noncanonical structures and proteins preferentially binding them often play a critical role in physiological molecular processes [32,104,105], but also in the progression of human diseases, such as various cancer types and neurodegenerative diseases, reviewed in [55,106,107]. Knowledge about the amino acid composition of various proteins binding noncanonical nucleic acids can be utilized as an additional clue/fingerprint in discovering novel noncanonical nucleic acid-binding protein candidates and therapeutically utilized [108,109,110].

The scheme below depicts sequence and structure-specific nucleic acid-binding phenomena in a nutshell (Figure 2).

Almost every year, multiple novel noncanonical nucleic acid-binding proteins are identified. This year was, for instance, found that Guanine Nucleotide-Binding Protein-Like 1 (GNL1) binds RNA G-quadruplex structures in genes associated with Parkinson’s disease [111], or that Small Nuclear Ribonucleoprotein Polypeptide A (SNRPA) directly binds to the BAG-1 mRNA through the G-quadruplex which can modulate BAG-1 expression level [112] (anti-apoptotic BAG-1 protein is known to be overexpressed in colorectal cancers [113]). Prediction of proteins that preferentially bind noncanonical DNA/RNA structures, therefore, should be a logical first step towards rapid identification of novel therapeutic targets for future treatment of severe human diseases.

## 3. Closing Remarks

The global or local amino acid composition of nucleic acid-binding proteins is often overlooked and an unjustly underestimated parameter. Mainly statistically significant enrichment or depletion of particular amino acid residues may serve as a promising tool to predict novel proteins with a similar function, as it was confirmed e.g., for G-quadruplex binding proteins.

## Figures and Tables

**Figure 1 ijms-22-00922-f001:**
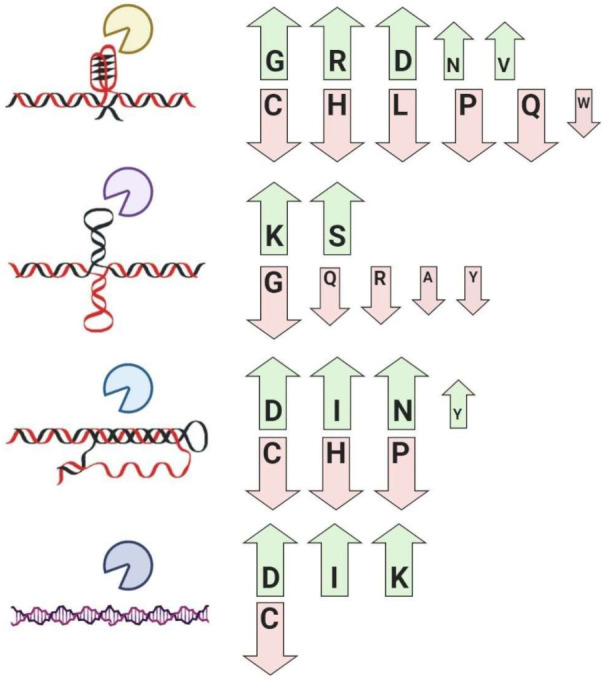
Significantly enriched and depleted amino acid residues (one letter aa code) in the dataset of G-quadruplex binding proteins (top), cruciform binding proteins, triplex binding proteins, and Z-DNA-binding proteins. Using Bonferroni correction, only values lower than 0.0025 were taken as significant (*p* <0.0025; *p* <0.0010; *p* <0.0001). The size of arrows indicates the significance of enrichment/depletion on scale (highest, moderate, lowest). Figure compiled using data from [47,48,89]. Created with BioRender.com.

**Figure 2 ijms-22-00922-f002:**
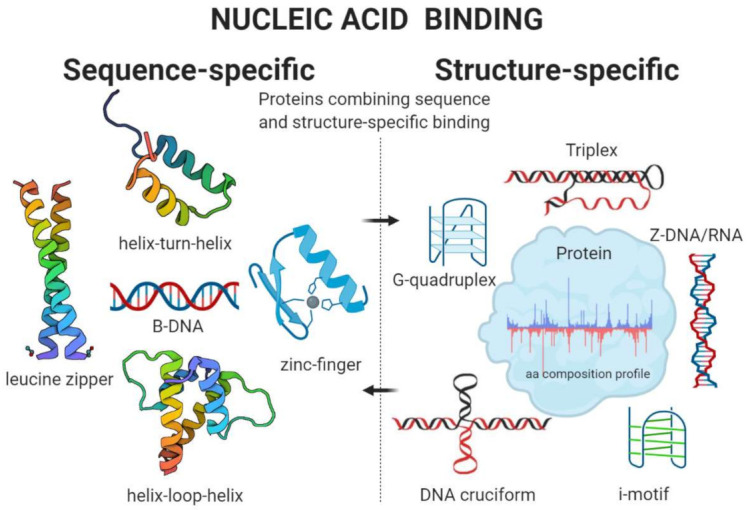
Types of nucleic acid-binding. The nucleic acid-binding mechanism can be basically divided into two main categories—sequence and structure-specific binding. (Left) Sequence-specific binding proteins recognize the variety of known DNA/RNA sequences via specific interaction with well-characterized protein motifs (zinc-fingers, helix-loop-helix, leucine zipper, helix-turn-helix, etc.). (Right) Structure-specific binding proteins recognize specific local structure(s) of nucleic acids, e.g., G-quadruplexes, i-motifs, cruciforms, triplexes, Z-DNA, and many others. In fact, it is a very common phenomenon that protein with the sequence-specific binding also prefers local DNA/RNA structure in its binding site or within the near neighborhood (e.g., p53), which is indicated by vertical black dashed line and arrows. Created with BioRender.com.

**Table 1 ijms-22-00922-t001:** Types of nucleic acid-binding proteins. This table summarizes the main categories of nucleic acid-binding proteins. There are two points of view. At first, we can simply divide these proteins into DNA and RNA-binding ones (and a relatively small category of proteins that are able to bind both DNA and RNA). Secondly (and more importantly), we can distinguish proteins that specifically bind known sequence motifs (sequence-specific DNA/RNA-binding) and proteins, which specifically bind local DNA/RNA structures. Besides, keep in mind that this table is very simplified, and categories are divided to be reader-friendly. In fact, many of the DNA/RNA-binding proteins combine sequence and structure-specific binding mechanisms.

	Important Notes	References
DNA-binding	Arginine, tryptophan, tyrosine, histidine, phenylalanine, and lysine residues enrichment. Glutamate, aspartate, and proline depletion in the protein-DNA interface.	[76,79]
RNA-binding	Arginine, methionine, histidine, and lysine residues enrichment. Glutamate, aspartate residues depletion in protein-RNA interface.	[75,76]
DNA and RNA-binding	Proteins that are able to bind both DNA and RNA.	[74,80]
**Sequence-specific**		
Zinc finger proteins	Cysteine and histidine amino acid residues are crucially important to coordinate Zn^2+^ binding in the Cys_2_His_2_ subgroup of zinc-finger proteins	[77,78]
Helix-turn-helix (HTH)	Conserved “shs” and “phs” patterns, where ‘s’ is a small residue, most frequently glycine in the first position, ‘h’ is a hydrophobic residue, and ‘p’ is a charged residue, most frequently glutamate. “shs” pattern lies in the turn between helix-2 and helix-3 of the core HTH structure, and “phs” is present in helix-2.	[81]
Basic Helix-loop-helix (bHLH)	Mostly arginine, lysine or histidine amino acid residues are present within conserved positions of this motif	[82,83]
Leucine zipper proteins	Leucine amino acid residues are crucial for leucine zipper motifs	[84,85]
**Structure specific**		
G-quadruplex binding proteins	Global enrichment for glycine, arginine, aspartic acid, asparagine, valine, and depletion for cysteine, histidine, leucine, proline, glutamine, and tryptophan residues	[47,86,87,88]
Cruciform binding proteins	Global enrichment for lysine and serine, and depletion for alanine, glycine, glutamine, arginine, tyrosine, and tryptophan residues	[48,55]
Triplex binding proteins	Global enrichment for asparagine, aspartic acid, isoleucine, tyrosine, and depletion for cysteine, histidine, and proline residues	[89]
Z-DNA/RNA-binding proteins	Global enrichment for isoleucine, aspartic acid, lysine, and depletion for cysteine residues	[89]

## Data Availability

No new data were created or analyzed in this study. Data sharing is not applicable to this article.

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
