# Peer review of "Amino Acid Composition in Various Types of Nucleic Acid-Binding Proteins"

_ijms, 2021, doi:10.3390/ijms22020922_

Round 1
Reviewer 1 Report
This manuscript is a review on nucleic-acid binding proteins, with an emphasis on the amino-acid composition biases (enrichment or depletion) by binding type (sequence or structure specific) and protein categories.
The review is fairly short, its aims are clear, and it reads well. The English language requires some editing (there are a number of missing articles, e.g. lines 51, 72, 188 twice, 196, and there a few syntax issues, e.g. the use of '[...], how to [...]' on lines 87 or 96), but nothing critical.
My few comments and questions are:
- It seems to me that the Figure 1 duplicates (partially) information already given in the Table 1 (the enrichment/depletion) and the Figure 2 (structural schemes). It only adds the significance of the enrichment/depletion (albeit not for all categories). Why not provide this information directly in the table, using e.g. ---, --, -, +, ++, +++ for large depletion to large enrichment?
- On what quantitative basis was the significance of enrichment/depletion categorized (i.e. what thresholds were used to rank into low/moderate/high)? I note that the data was compiled from the authors' own studies; was this kind of analysis performed by others (possibly proteins not covered by the authors). If yes, are they using the same categorization thresholds, and can their results be added to the table?
- Are these enrichments/depletions in amino-acid particularly significant at the protein binding sites, and if yes could the author discuss it?
- In the same vein, is it known if these composition biases play a role in the binding to any given structure (in terms of molecular interactions)?
- Could the author please clarify the sentence line 200. What is meant by 'virtually multiple times a year'?
- It may be worth annotating the non-B structures from the right panel of figure 2 (the same way it was done in the left panel)
Overall, I recommend publication of this manuscript with minor revisions.
Author Response
This manuscript is a review on nucleic-acid binding proteins, with an emphasis on the amino-acid composition biases (enrichment or depletion) by binding type (sequence or structure specific) and protein categories.
The review is fairly short, its aims are clear, and it reads well. The English language requires some editing (there are a number of missing articles, e.g. lines 51, 72, 188 twice, 196, and there a few syntax issues, e.g. the use of '[...], how to [...]' on lines 87 or 96), but nothing critical.
Dear reviewer, thank you very much for carefully reading our manuscript and for all the comments and useful suggestions. We have answered them all below and carried out additional professional English proofreading (please see the tracked-changes).
My few comments and questions are:
1. It seems to me that the Figure 1 duplicates (partially) information already given in the Table 1 (the enrichment/depletion) and the Figure 2 (structural schemes). It only adds the significance of the enrichment/depletion (albeit not for all categories). Why not provide this information directly in the table, using e.g. ---, --, -, +, ++, +++ for large depletion to large enrichment?
Thanks for the comment and of course, we agree with you, that the information in Table and Figure is partially duplicated. On the other side, we made this figure to serve as a take-home message for readers, because it is much more well-arranged than a Table (where the purpose was mainly to made a clear division between nucleic acid-binding proteins, and comprise and sort all references). So, we would like to preserve both Table and Figure, if you don’t mind.
2. On what quantitative basis was the significance of enrichment/depletion categorized (i.e. what thresholds were used to rank into low/moderate/high)? I note that the data was compiled from the authors' own studies; was this kind of analysis performed by others (possibly proteins not covered by the authors). If yes, are they using the same categorization thresholds, and can their results be added to the table?
Thanks for the comment, to our best knowledge, described type of analyses (amino acid composition of proteins that preferentially bind to Non-B DNA) were still performed only in our own previously published studies. We added the better explanation of quantitative basis of significance to the Figure legend.
3. Are these enrichments/depletions in amino-acid particularly significant at the protein binding sites, and if yes could the author discuss it?
Thanks for the comment, it would be logical and we believe that yes. Unfortunately, for the vast majority of analysed Non-B DNA binding proteins, the knowledge about exact Non-B DNA binding site(s) is still missing. We have added a sentence about it in the Discussion.
4. In the same vein, is it known if these composition biases play a role in the binding to any given structure (in terms of molecular interactions)?
Thanks for the comment, it becomes more and more evident, that amino acid compositional biases play an important role in protein-protein interactions, see e.g. https://www.sciencedirect.com/science/article/pii/S0022283619305145. On the other hand, the role of aa biases in Non-B DNA – protein interaction is the very new concept, first proposed in our mini-review.
5. Could the author please clarify the sentence line 200. What is meant by 'virtually multiple times a year'?
Thank you for your comment, we have reformulated this sentence to be more accurate.
6. It may be worth annotating the non-B structures from the right panel of figure 2 (the same way it was done in the left panel)
Thank you for the comment, we have done a new version of the Figure 2.
Overall, I recommend publication of this manuscript with minor revisions.
Reviewer 2 Report
Review is well-written and very clear. I have no major concerns, even if for completness about the interaction between proteins and non-canonical structures as G-quadruplexes, the following references should be added to line 59: 1) doi: 10.1016/j.ijbiomac.2019.04.141 and 2) doi: 10.1002/med.21737.
Author Response
Review is well-written and very clear. I have no major concerns, even if for completness about the interaction between proteins and non-canonical structures as G-quadruplexes, the following references should be added to line 59: 1) doi: 10.1016/j.ijbiomac.2019.04.141 and 2) doi: 10.1002/med.21737.
Dear reviewer, thank you very much for your positive feedback and useful suggestion about missing references. We have added them and highlighted them in yellow.